# Thermodynamic efficiency in dissipative chemistry

Emanuele Penocchio [1], Riccardo Rao [1,2] & Massimiliano Esposito[1]

Chemical processes in closed systems inevitably relax to equilibrium. Living systems avoid this fate and give rise to a much richer diversity of phenomena by operating under nonequilibrium conditions. Recent experiments in dissipative self-assembly also demonstrated that by opening reaction vessels and steering certain concentrations, an ocean of opportunities for artificial synthesis and energy storage emerges. To navigate it, thermodynamic notions of energy, work and dissipation must be established for these open chemical systems. Here, we do so by building upon recent theoretical advances in nonequilibrium statistical physics. As a central outcome, we show how to quantify the efficiency of such chemical operations and lay the foundation for performance analysis of any dissipative chemical process.

[1] Complex Systems and Statistical Mechanics, Physics and Materials Science Research Unit, University of Luxembourg, L-1511 Luxembourg, Luxembourg. [2] Present address: The Simons Center for Systems Biology, School of Natural Sciences, Institute for Advanced Study, Princeton, NJ 08540, USA. Correspondence and requests for materials should be addressed to M.E. (email: massimiliano.esposito@uni.lu)

Traditional chemical thermodynamics deals with closed systems, which always evolve towards equilibrium. At equilibrium, all reaction currents—defined as the forward reaction fluxes minus the backwards ($J_\rho = J_{+\rho} - J_{-\rho}$, where $\rho$ labels the reactions)—eventually vanish. The first thermodynamic description of nonequilibrium chemical processes was achieved by the Brussels school founded by de Donder and perpetuated by Prigogine[1,2], but they focused on few reactions close to equilibrium in the so-called linear regime. However, processes such as fuel-driven self-assembly involve open chemical reaction networks (CRN) with many reactions operating far away from equilibrium[3,4]. The openness arises from the presence of one or more chemostats, i.e. particle reservoirs coupled with the system which externally control the concentrations of some species—just like thermostats control temperatures—and allow for matter exchanges. Open CRN can then be thought of as thermodynamic machines powered by chemostats. Two operating regimes may be distinguished, reminiscent of stroke and steady-state engines. In the first, work is used to induce a time-dependent change in the species abundances that could never be reached at equilibrium. An example could be the accumulation of a large amount of molecules with a high free energy content as in fuel-driven self-assembly, or the depletion of some undesired species as in metabolite repair[5]. In the second, work is used to maintain the system in a nonequilibrium stationary state which continuously transduces an input work into useful output work. Beyond energy transduction within pseudo-first order reactions[6], no framework currently exists to assess how efficient and powerful such chemical engines can be. We provide one grounded in the recently established nonequilibrium thermodynamics of CRN[7,8], which was born from the combination of state-of-the-art statistical mechanics[9–14] and mathematical CRN theory[15,16]. Establishing rigorous concepts of free energy, chemical work and dissipation valid far from equilibrium reveals crucial. They provide the basis for thermodynamically meaningful definitions of efficiencies and optimal performance in the different operating regimes. In the following, energy storage (ES) and driven synthesis (DS) are analyzed as models epitomizing the first and the second operating regime, respectively, but our findings apply to any dissipative chemical process.

## Results

**Energy storage**. In energy storage, an open CRN initially at equilibrium with high concentrations of low-energy molecules and low concentrations of high-energy ones is brought out of equilibrium with the aim to increase the concentrations of the high-energy species. This process is reminiscent of charging a capacitor via the coupling to a voltage generator. In the context of supramolecular chemistry, the concept of ES was proposed by Ragazzon and Prins[4]. An insightful model capturing its main features is described in Fig. 1. Its thermodynamic analysis, detailed in Supplementary Note 1b, will now be outlined. Given a set of reaction rate constants, the accumulation of the high-energy species $A_2$ may be enabled when chemostats set a certain positive chemical potential difference of fuel and waste, i.e. $\mathcal{F}_{\text{fuel}} = \mu_F - \mu_W > 0$, by steering [F] (see Supplementary Fig. 1). This implies the injection of F molecules at a rate $I_F$ and the extraction of W at rate $I_W$. The resulting power (i.e., work per unit of time) performed on the system by the fueling mechanism is $\dot{\mathcal{W}}_{\text{fuel}} = I_F \mathcal{F}_{\text{fuel}}$[7,8,17]. The proper way to quantify the energy content of an open CRN is via its nonequilibrium free energy $\mathcal{G}$. During the charging process, only part of the work, namely $\Delta\mathcal{G}$, is dedicated to shift the concentrations distribution and is stored as free energy in the system[4]. The remaining fraction, namely $T\Sigma$, is dissipated according to the

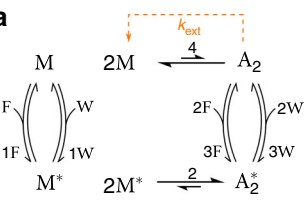

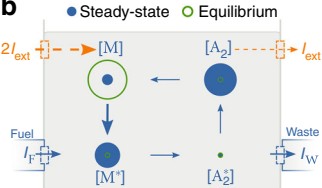

**Fig. 1** Model for energy storage and driven synthesis. Without (resp. with) the orange dashed transition, the chemical reaction network models energy storage (resp. driven synthesis). The high-energy species $A_2$ is at low concentration at equilibrium. Powering the system by chemostatting fuel (F) and waste (W) species boosts the formation of $A_2$ out of the monomer M via the activated species $M_2$ and $A_2^*$. **a** The chemical reaction network (forward fluxes are defined counter-clockwise). **b** Sketch of concentrations distributions (proportional to radii) and net currents (proportional to arrows thickness, see Supplementary Note 1c)

second law of thermodynamics

$$\mathcal{W}_{\text{fuel}} = \Delta\mathcal{G} + T\Sigma, \tag{1}$$

where $T$ is temperature and $\Sigma \geq 0$ the entropy production, which only vanishes at equilibrium. The time-dependent thermodynamic efficiency of an ES process is thus the ratio

$$\eta_{\text{es}} = \frac{\Delta\mathcal{G}}{\mathcal{W}_{\text{fuel}}} = 1 - \frac{T\Sigma}{\mathcal{W}_{\text{fuel}}}. \tag{2}$$

Equation (1) has been used to derive the second equality. We emphasize that each of these contributions has an explicit expression in terms of concentrations and rate constants (see Supplementary Note 1b). For instance, the energy stored at any time with respect to equilibrium is given by the expression

$$\Delta\mathcal{G} = RT \sum_{\substack{X=M, \\ M^*, A_2^*, A_2}} \left[ [X] \ln \frac{[X]}{[X]_{\text{eq}}} - [X] + [X]_{\text{eq}} \right] \geq 0, \tag{3}$$

which is reminiscent of an information theoretical measure called relative entropy[18]. Crucially, any concentration distribution different from the equilibrium one has a positive energy content. Equation (1) thus implies that an amount of work of at least $\Delta\mathcal{G}$ needs to be provided to reach it. It also ensures that $\eta_{\text{es}}$ is bounded between zero and one.

We simulated an ES process and plotted the dynamics of concentrations as well as efficiency and its contributions in Fig. 2. The process can be divided into a charging and a maintenance phase. During the former, the system energy grows ($d_t\mathcal{G} > 0$) in a way which correlates with the accumulation of the high-energy species $A_2$. The process can be quite efficient as a significant portion of the work is converted into free energy. However, in the maintenance phase, the system has reached a nonequilibrium steady state. The efficiency drops towards zero (proportional to the inverse time) as the entire work is being spent to preserve the energy previously accumulated ($d_t\mathcal{G} \simeq 0$). The maximum $\eta_{\text{es}}$ is reached during the charging phase (see Supplementary Note 1b for a rigorous proof) and defines

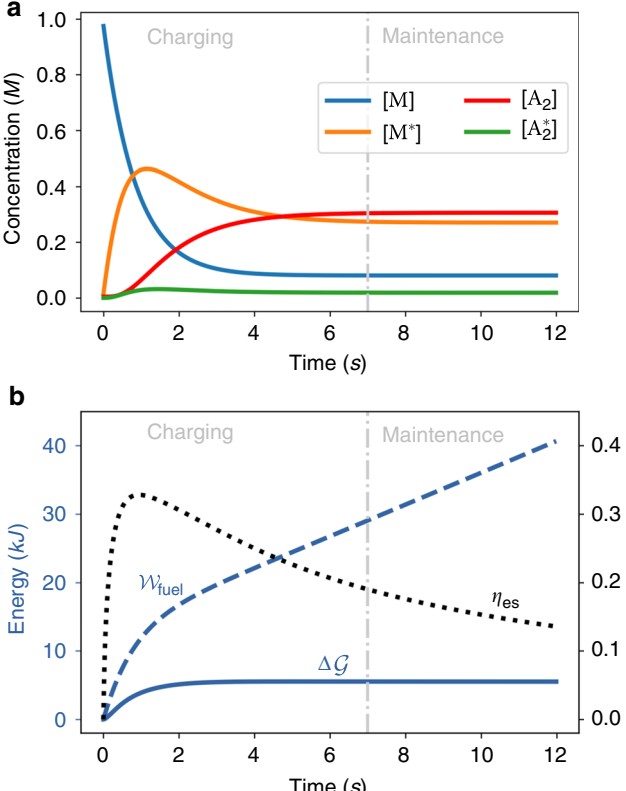

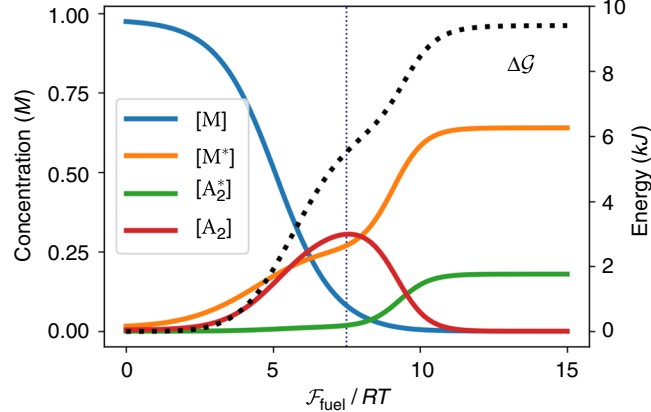

**Fig. 3** Maintenance phase of energy storage. Stationary concentrations and free energy difference from equilibrium in the maintenance phase of energy storage as a function of the chemical potential difference between fuel and waste. The vertical dotted line denotes the value $\mathcal{F}_{\text{fuel}} = 7.5 \cdot RT$ used to study the charging phase in Fig. 2

**Fig. 2** Dynamics of energy storage. The system is initially prepared at thermodynamic equilibrium where $[\text{M}]_{\text{eq}} \gg [\text{A}_2]_{\text{eq}}$. At time $t = 0$, the chemical potential difference between fuel and waste is turned on at $\mathcal{F}_{\text{fuel}} = 7.5 \cdot RT$ and drives the system away from equilibrium. After a transient (charging phase), the system eventually settles into a nonequilibrium steady state (maintenance phase). **a** Species abundances. **b** Energy stored, work and efficiency (right axis, adimensional units). Kinetic constants ($\{k_{\pm\rho}\}$) and chemical potentials used for simulations are given in Supplementary Table 1

the time that minimizes the dissipation of ES. The value of the efficiency when the process enters the maintenance phase characterizes instead the performance of the ES process when the system has reached its maximum storage capacity. The best time to stop ES and start making use of it (cf. driven synthesis below) will be a tradeoff between maximizing the energy stored and minimizing dissipation. In general, the ideal situation will be the one in which $\eta_{\text{es}}$ peaks as close as possible to the maintenance phase.

Figure 3 focuses on the maintenance phase for different values of $\mathcal{F}_{\text{fuel}}$. It shows that by driving the system away from equilibrium, one can reach species abundances that are very different with respect to the equilibrium ones. It also shows that the accumulation of free energy does not necessarily coincide with an increase in concentration of the most energetic species $\text{A}_2$. Indeed, while at low values of $\mathcal{F}_{\text{fuel}}$ the accumulation of $\mathcal{G}$ correlates with $[\text{A}_2]$, beyond a threshold $\text{A}_2$ starts to be depleted while energy continues getting stored by further moving away the concentration distribution from equilibrium. We finally note that the connection of our work to "kinetic asymmetry"[4,19] is discussed in Supplementary Note 1c.

As we have seen, the crucial part of energy storage is the charging phase, as the maintenance phase is purely dissipative and consumes chemical work without any energy gain. In order to make use of the energy accumulated during the charging phase, a mechanism extracting the energetic species from the system

must be introduced. This complementary but distinct working regime of an open CRN will now be considered.

**Driven synthesis**. In driven synthesis, an energetic species that accumulates thanks to a fueling process is continuously extracted from a system in a nonequilibrium steady state. One may consider for instance processes where the product either evaporates, precipitates or undergoes other fast transformations while being rapidly replaced by reactants. By building upon the above ES scheme, a simple way to model DS is to add an ideal extraction/injection mechanism to the CRN (orange dashed arrows in Fig. 1). This mechanism removes the assembled molecule $\text{A}_2$ and renews two M molecules at a rate $I_{\text{ext}} = k_{\text{ext}}[\text{A}_2]$. In doing so, we model the endergonic synthesis of molecules that are strongly unfavored at equilibrium, a strategy used by Nature[20–22] and which may be within reach of supramolecular chemists[23–25].

From the thermodynamic standpoint detailed in Supplementary Note 2b, the input power spent by the fueling mechanism, $\dot{\mathcal{W}}_{\text{fuel}} = I_{\text{F}}\mathcal{F}_{\text{fuel}} = I_{\text{F}}(\mu_{\text{F}} - \mu_{\text{W}})$, is now not just dissipated as $T\dot{\Sigma}$, but part of it is used to sustain the production of $\text{A}_2$:

$$\dot{\mathcal{W}}_{\text{fuel}} = -\dot{\mathcal{W}}_{\text{ext}} + T\dot{\Sigma}. \tag{4}$$

The output power released by the extraction mechanism, $\dot{\mathcal{W}}_{\text{ext}} = I_{\text{ext}}(2\mu_{\text{M}} - \mu_{\text{A}_2})$, is negative when DS occurs. In this context the thermodynamic efficiency is thus given by

$$\eta_{\text{ds}} = -\frac{\dot{\mathcal{W}}_{\text{ext}}}{\dot{\mathcal{W}}_{\text{fuel}}} = 1 - \frac{T\dot{\Sigma}}{\dot{\mathcal{W}}_{\text{fuel}}}, \tag{5}$$

where Eq. (4) has been used to derive the second equality. It is bounded between zero and one when DS occurs.

In Fig. 4, we simulated DS for various working conditions by varying $k_{\text{ext}}$ and $\mathcal{F}_{\text{fuel}}$. We start our analysis by considering a given value of $\mathcal{F}_{\text{fuel}}$. As $k_{\text{ext}}$ is increased, $\eta_{\text{ds}}$ first grows to an optimal value before decreasing towards negative values where the DS regime ends (see Fig. 4a). At the same time $I_{\text{ext}}$ increases until it reaches a plateau (see Fig. 4c). This happens when $k_{\text{ext}}$ overcomes the ability of the system to sustain high values of $[\text{A}_2]$ (Fig. 4b). Eventually the drop in $[\text{A}_2]$ is such that $2\mu_{\text{M}} - \mu_{\text{A}_2} > 0$, thus resulting in the loss of the DS regime. We now fix $k_{\text{ext}}$ and increase $\mathcal{F}_{\text{fuel}}$. The DS regime starts at a threshold value, when $[\text{A}_2]$ becomes high enough. After that, both $[\text{A}_2]$ and the efficiency grow to an optimal value before decreasing again. This time however, the efficiency remains positive as $[\text{M}]$ drops

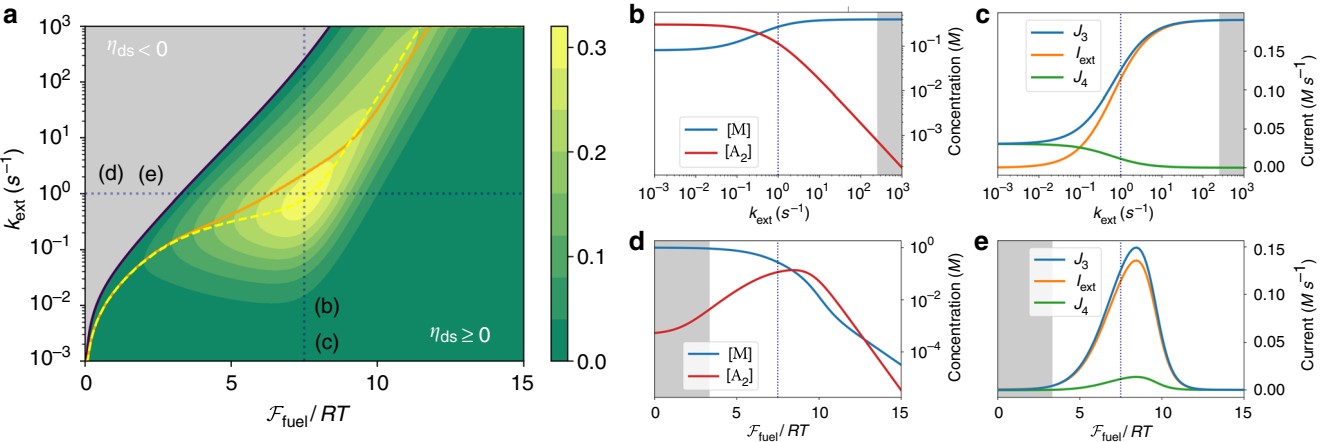

**Fig. 4** Performance of driven synthesis. **a** Efficiency ($\eta_{ds}$) of driven synthesis as function of $\mathcal{F}_{fuel}$ and $k_{ext}$. Regions of operating regimes that do not perform driven synthesis are colored in gray. The yellow dashed (resp. orange solid) line denotes the maximum of $\eta_{ds}$ (resp. $-\dot{W}_{ext}$, see also Supplementary Fig. 2) versus $k_{ext}$, while the black solid line corresponds to $\eta_{ds} = 0$. **b, c** (resp. **d, e**) Concentrations and currents as a function of $k_{ext}$ (resp. $\mathcal{F}_{fuel}$) along the line $\mathcal{F}_{fuel} = 7.5 \cdot RT$ (resp. $k_{ext} = 1\,s^{-1}$) highlighted by blue vertical (resp. horizontal) dotted line in plot (**a**). Kinetic constants and standard chemical potentials are the same as for ES analysis (see Supplementary Information). Note that $J_3 = I_{ext} + J_4$ always holds, where $J_3 = J_{3F} + J_{3W}$ is the net current flowing from $A_2^*$ to $A_2$

together with [A₂] (see Fig. 4d). Figure 4e shows another important feature. As $\mathcal{F}_{fuel}$ is increased, $I_{ext}$ first increases too, but eventually reaches a maximum after which it decreases. This phenomenon is a hallmark of far-from-equilibrium physics which could not happen in a linear regime, namely when $k_{ext}$ and $\mathcal{F}_{fuel}$ are small. Remarkably, the global maximum of the efficiency in Fig. 4a is reached in a region far from equilibrium. We note that it corresponds to values of $\mathcal{F}_{fuel}$ close to the one maximizing [A₂] in the maintenance phase of ES (see Fig. 3) and to values of $k_{ext}$ of order one resulting in $I_{ext}$ which do not overly deplete [A₂]. We finally turn to the lines of maximum efficiency and efficiency at maximum power in Fig. 4a, where the maximization is done with respect to $k_{ext}$ at a given $\mathcal{F}_{fuel}$. Since these two lines typically do not coincide, the study of the tradeoffs is the object of a rich field called finite-time thermodynamics[26]. Interestingly, while these two lines cannot coincide in the linear regime (see Supplementary Note 2d), we see that they do intersect far-from-equilibrium, not far from the global maximum of the efficiency. Our analysis thus allowed us to identify a region of good tradeoff between power and efficiency for the model of DS we introduced. In order to emphasize the fact that all the interesting features that we identified in DS occur far-from-equilibrium, we analyze in detail in Supplementary Note 2d the linear regime of DS. After identifying the Onsager matrix, we are able to analytically reproduce the results of the simulations in the limit of small $\mathcal{F}_{fuel}$ and $k_{ext}$ (bottom-left part of Fig.4a, see Supplementary Fig. 3 for details), thus pinpointing the limit of validity of the linear regime approximation.

## Discussion
Thermodynamics was born from the effort to systematize the performance of steam engines. Open CRN, which are at the core of recent efforts in artificial synthesis[27] and ubiquitous in living systems[22,28,29], can be seen as chemical engines. In the spirit of this analogy, in this article we built a chemical thermodynamic framework which enables us to systematically analyze the performance of two fundamental dissipative chemical processes. The first, energy storage, is concerned with the time-dependent accumulation of high-energy species far from equilibrium and is currently raising significant attention from supramolecular chemists. The second, driven synthesis, aims at continuously extracting the

previously obtained high-energy species and provides a simple and insightful instance of energy transduction beyond pseudo-unimolecular CRN. In doing so, we identified their optimal regimes of operation. Crucially they lie far-from-equilibrium in regions unreachable using conventional linear regime thermo-dynamics. We emphasize that the methods developed in this paper can in principle be applied to any open CRN and thus provide the basis for future performance studies and optimal design of dissipative chemistry. They are thus destined to play a major role in bioengineering and nanotechnologies.

## Data availability
All data needed to reproduce numerical results are reported in the Supplementary Information.

## Code availability
The code that generated the plots is available from the corresponding author upon request.

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

## Acknowledgements

This work was funded by the Luxembourg National Research Fund (AFR Ph.D. Grant 2014-2, No. 9114110) and the European Research Council project NanoThermo (ERC-2015-CoG Agreement No. 681456).

## Author contributions

E.P., R.R., and M.E. all significantly contributed to conceive and realize the project as well as in writing the paper.

## Additional information

**Competing interests:** The authors declare no competing interests.

