## [Peer Review File · Nature Communications]

Reviewers' comments:

Reviewer #1 (Remarks to the Author):

The work of Esposito et al. describes the thermodynamic efficiency of a model chemical reaction network. The model network is of particular interest because it is the minimal model associated to microtubules formation and has considerable chances to be implemented by experimental chemists in fully synthetic systems.

The work focuses on how the external energy source is exploited by the reactive network, i.e. how efficiently the system absorbs energy from the driving reaction. The authors identify two phases, one in which energy is absorbed from the driving reaction, and the other in which energy is completely dissipated to maintain a nonequilibrium distribution. The analysis focuses on the processes of storing energy and exploiting it to obtain a high-energy target species, that is then removed (i.e. dissipative synthesis).

Crucially, the author show how to analyze and quantify thermodynamic efficiency in a regime "far from equilibrium". This is highly interesting, since it is generally accepted that most of nonequilibrium chemical processes (and the associated interesting phenomena) occur outside the linear regime, i.e. "far from equilibrium".

An outstanding quality of this manuscript is that the authors successfully attempted to describe their results in a way that is accessible also to experimental chemists. At present, there are at least two communities that are working on this subject (experimental chemist and theoretical physicists - with a rough simplification) but these communities most of the times do not communicate, and this strongly hampers the development of the field. Here the authors could convey their message in a way that can be understood and appreciated by experimental chemists, and the broad readership of Nat. Commun..

Overall the work is highly interesting and rather unique in the framework of theoretical investigations on nonequilibrium systems.

Some suggestions follow, to further improve the quality of the work:

Main issues.

(1) As anticipated, the chosen chemical reaction scheme is very interesting, since it reflects how Nature operates, and how supramolecular chemists might be able to drive an assembly of artificial molecules, however it has some unique features that would be better described in the text. The authors purposely introduced a kinetic asymmetry in the system, and --crucially- this governs the overall behavior, i.e. the accumulation of A₂ is intrinsic in the kinetic constants used and is not a general behavior;

(1.1) To facilitate the comprehension of chemists interested in the topic, it would be desirable to report in the SI also the backward kinetic constants and the associated equilibrium constants. At the present stage it is very difficult for an experimental chemist to understand the kinetic choices at the basis of the model (that are critical for its operation). A similar comment applies to the concentration of Fuel: it would be desirable to explicitly understand in which concentration range the fuel is varying. At the moment the model looks almost like a black box, at first sight;

(2) Partly connected to the previous points: for the experimental community it is critical to understand how self-assembly equilibria are affected by fuel consumption, indeed the treatments on kinetic asymmetry focus purposely only on those equilibria. Here the reactions with fuel and waste are also considered to calculate the overall efficiency. Pointing out this difference would benefit the work, as it would clarify what is being calculated, and would put this work in the right perspective with the works of Astumian and Prins on kinetic asymmetry;

(2.1) It would be interesting, if possible/appropriate, to understand if the overall efficiency has a component associated to the self-assembly equilibria, and a second component associated to the interactions of Fuel and Waste;

Overall point 1 and 2 would show that/how the present manuscript and the considerations on kinetic asymmetry are indeed complementary features, and this is an important message that creates a common ground between different approaches to the same topic;

(3) The authors are encouraged to carefully consider the use of "dissipative". Dissipation is the conversion of energy into heat in an irrecoverable way, and the word "dissipative" is used by chemist also as a broad term comprising the chemical processes that lead to heat dissipation. In some cases the authors mean "driven"/"endergonic" and I encourage the authors to use the more specific term whenever possible. For example, at line 10 of the main text "dissipative" seems the appropriate term (fine also in the title), whereas e.g. at line 24 "driven" would be more appropriate

than "dissipative". In the chemical literature there is some confusion in the use of this terminology, therefore it is desirable to avoid further misunderstandings;

(4) A similar comment applies to the use of "dissipative synthesis", that -to the best of my knowledge- is a neologism of the authors in this context. I encourage the authors to carefully consider if this is necessary or not. One alternative could be to use "endergonic synthesis", since an endergonic reaction is taking place, or "driven synthesis", that would be less technical, but would match the use of "driven self-assembly". In the humble opinion of this reviewer the use of "endergonic synthesis" could also potentially attract the interest of (and inspire!) synthetic chemists, as they are well aware of many strategies to perform endergonic reactions;

Minor issues, aiming at an improved readability:

(5) In the SI there is likely a mistake in (S1), in the flux matrix, in the rates associated to step 3. Same for S2. The matrix is instead the correct one in S22;

(6) Considering the broad audience at which Nat. Commun. aims, I would avoid stating that "closed systems are ""poorly"" controllable", since in the laboratory practice there is a fair amount of control on closed systems, and it is extremely difficult to control open systems. I guess that the authors mean that there is limited room for controlling them. Stating that they are "highly predictable" could be one option to avoid potential misunderstandings.

(7) In figure 1b, (a) it is not clear to the reader why the dashed arrow pointing at [M] is thicker than the corresponding one going away from [A2]; (b) it is not clear why the vertical arrows on the left are thicker and the equilibrium is shifted towards [M*], whereas in the right arrows they are shifted towards [A2]. They are not reflecting thermodynamic stability, but neither the populations under dissipative conditions (as they would be in conflict with the horizontal arrows). The authors are encouraged to reconsider this graphical aspects (if appropriate);

(8) In the text associated to figure 4 the authors are encouraged to explicitly refer in the text to the dotted lines (b) and (c), because at first it is difficult to relate the description made in the text to a specific point of fig. 4a. Making the dotted lines more evident (extended) in fig 4 a may also improve the readability of the figure and the associated text. Also, it would be advisable to change/adjust the label of figures (b) and (c), as the concentration panels seem not referenced at first; this would help also the referencing to the concentration panels in the main text;

(9) Please be more specific when stating that it was possible to reproduce "bottom left part of figure 4a";

(10) After eq. 2, please consider if "inequality" is appropriate (equality?);

(11) At the end of page 2, please consider if "reaches" is appropriate (reached?);

(12) Please consider if the use of ""burns" chemical work" at page 3 is appropriate.

Reviewer #2 (Remarks to the Author):

In this paper the authors seek to develop a thermodynamic framework for analyzing the performance of chemical reaction networks (CRN) that accomplish either of two useful tasks: the accumulation of high energy molecules (energy storage, or ES), or the extraction of these molecules (dissipative synthesis, or DS).

In either case, the performance is characterized by efficiency, defined to be the ratio of a desired output to an expended input. For ES, the output is the nonequilibrium free energy [curly G], which begins at zero in equilibrium and grows until it reaches a plateau as the concentration of high energy molecules saturates. The input is chemical work, calculated for the reaction (in this case, $F \rightarrow W$) that generates and then sustains the nonequilibrium steady state. Since [curly G] saturates while chemical work grows linearly (asymptotically) with time, the efficiency decays to zero with time, which simply reflects that once the steady state is reached, the continual reaction $F \rightarrow W$ brings no added benefits, but merely maintains a nonequilibrium concentration of high energy molecules.

For DS, the output is the combined extraction of high energy molecules and injection of low energy ones, and the input is again chemical work. This situation is an example of free energy transduction, and efficiency is thus naturally defined in terms of output chemical work over input chemical work.

The analysis in this paper is correct, and the definition of efficiency in the ES case seems novel. But since this definition gives an efficiency that decays with time, it is not clear why this definition is a good one or why we should expect it to influence thinking in the field. Yes, η_{es} (equation 2) has a

value between 0 and 1 in the ES regime, but why should we expect this to be a useful measure of performance?

In the DS case the definition of efficiency η_{ds} (equation 5) is quite reasonable, but in this case the efficiency is simply a ratio of output chemical work to input chemical work. This would seem to be a more or less "obvious" definition of efficiency for free energy transduction, which is the situation in which a downhill chemical reaction drives another reaction uphill. Admittedly, I can provide no reference where I have seen this definition used.

In short, while I find the analysis to be correct, I am not entirely convinced that the authors have developed a novel formalism that is likely to prove useful in the performance analysis of CRNs. To make a stronger case, the authors should address the following questions. For ES, how is their definition of efficiency useful, particularly as it decays to zero with time? For DS, does their definition differ from the ratio of output chemical work to input chemical work, in a straightforward example of free energy transduction?

**Reply to referee report for
*Thermodynamic Efficiency in Dissipative Chemistry***

We sincerely thank both referees for their comments and suggestions which gave us the opportunity to improve the quality and clarity of our manuscript. For convenience, below we reproduce the entire reports and reply in red to the points raised. We used the same color to highlight modifications in the revised version of the manuscript (main text + SI).

Reviewer #1 (Remarks to the Author):

The work of Esposito et al. describes the thermodynamic efficiency of a model chemical reaction network. The model network is of particular interest because it is the minimal model associated to microtubules formation and has considerable chances to be implemented by experimental chemists in fully synthetic systems.

The work focuses on how the external energy source is exploited by the reactive network, i.e. how efficiently the system absorbs energy from the driving reaction. The authors identify two phases, one in which energy is absorbed from the driving reaction, and the other in which energy is completely dissipated to maintain a nonequilibrium distribution. The analysis focuses on the processes of storing energy and exploiting it to obtain a high-energy target species, that is then removed (i.e. dissipative synthesis).

Crucially, the author show how to analyze and quantify thermodynamic efficiency in a regime "far from equilibrium". This is highly interesting, since it is generally accepted that most of nonequilibrium chemical processes (and the associated interesting phenomena) occur outside the linear regime, i.e. "far from equilibrium".

An outstanding quality of this manuscript is that the authors successfully attempted to describe their results in a way that is accessible also to experimental chemists. At present, there are at least two communities that are working on this subject (experimental chemist and theoretical physicists - with a rough simplification) but these communities most of the times do not communicate, and this strongly hampers the development of the field. Here the authors could convey their message in a way that can be understood and appreciated by experimental chemists, and the broad readership of Nat. Commun..

Overall the work is highly interesting and rather unique in the framework of theoretical investigations on nonequilibrium systems.

We greatly appreciate that the referee deems our work valuable, in particular for the experimental community.

Some suggestions follow, to further improve the quality of the work:

Main issues.

(1) As anticipated, the chosen chemical reaction scheme is very interesting, since it reflects how Nature operates, and how supramolecular chemists might be able to drive an assembly of artificial molecules, however it has some unique features that would be better described in the text. The authors purposely introduced a kinetic asymmetry in the system, and --crucially- this governs the overall behavior, i.e. the accumulation of A2 is intrinsic in the kinetic constants used and is not a general behavior;

A entire section has been added in the supplementary information (and a link to it in the paper) to precisely connect the concept of kinetic symmetry/asymmetry to our work.

(1.1) To facilitate the comprehension of chemists interested in the topic, it would be desirable to report in the SI also the backward kinetic constants and the associated equilibrium constants. At the present stage it is very difficult for an experimental chemist to understand the kinetic choices at the basis of the model (that are critical for its operation). A similar comment applies to the concentration of Fuel: it would be desirable to explicitly understand in which concentration range the fuel is varying. At the moment the model looks almost like a black box, at first sight;

We updated section Id of the Supplementary Information with all the requested data.

(2) Partly connected to the previous points: for the experimental community it is critical to understand how self-assembly equilibria are affected by fuel consumption, indeed the treatments on kinetic asymmetry focus purposely only on those equilibria. Here the reactions with fuel and waste are also considered to calculate the overall efficiency. Pointing out this difference would benefit the work, as it would clarify what is being calculated, and would put this work in the right perspective with the works of Astumian and Prins on kinetic asymmetry;

The new paragraph below Eq. (3) in the paper and the aforementioned new section in the SI address this point.

(2.1) It would be interesting, if possible/appropriate, to understand if the overall efficiency has a component associated to the self-assembly equilibria, and a second component associated to the interactions of Fuel and Waste;

As can be seen from the structure of eq. (3), our definition of energy stored (and consequently of efficiency) is formulated in terms of the displacements of the concentration of each species from their equilibrium values. While it is straightforward to single out the components associated with each species, there is no such distinction at the level of reactions.

Overall point 1 and 2 would show that/how the present manuscript and the considerations on kinetic asymmetry are indeed complementary features, and this is an important message that creates a common ground between different approaches to the same topic;

(3) The authors are encouraged to carefully consider the use of "dissipative". Dissipation is the conversion of energy into heat in an irrecoverable way, and the word "dissipative" is used by chemist also as a broad term comprising the chemical processes that lead to heat dissipation. In some cases the authors mean "driven"/"endergonic" and I encourage the authors to use the more specific term whenever possible. For example, at line 10 of the main text "dissipative" seems the appropriate term (fine also in the title), whereas e.g. at line 24 "driven" would be more appropriate than "dissipative". In the chemical literature there is some confusion in the use of this terminology, therefore it is desirable to avoid further misunderstandings;

We followed the suggestions from the referee.

(4) A similar comment applies to the use of "dissipative synthesis", that -to the best of my knowledge- is a neologism of the authors in this context. I encourage the authors to carefully consider if this is necessary or not. One alternative could be to use "endergonic synthesis", since an endergonic reaction is taking place, or "driven synthesis", that would be less technical, but would match the use of "driven self-assembly". In the humble opinion of this reviewer the use of "endergonic synthesis" could also potentially attract the interest of (and inspire!) synthetic chemists, as they are well aware of many strategies to perform endergonic reactions;

We followed the suggestions from the referee and opted for "driven synthesis". We also made reference to endergonic reactions in the main text.

Minor issues, aiming at an improved readability:

(5) In the SI there is likely a mistake in (S1), in the flux matrix, in the rates associated to step 3. Same for S2. The matrix is instead the correct one in S22;

We corrected the mistake.

(6) Considering the broad audience at which Nat. Commun. aims, I would avoid stating that "closed systems are "poorly" controllable", since in the laboratory practice there is a fair amount of control on closed systems, and it is extremely difficult to control open systems. I guess that the authors mean that there is limited room for controlling them. Stating that they are "highly predictable" could be one option to avoid potential misunderstandings.

We changed the abstract to avoid the use of "poorly controllable".

(7) In figure 1b, (a) it is not clear to the reader why the dashed arrow pointing at [M] is thicker than the corresponding one going away from [A2]; (b) it is not clear why the vertical arrows on the left are thicker and the equilibrium is shifted towards [M*], whereas in the right arrows they are shifted towards [A2]. They are not reflecting thermodynamic stability, but neither the populations under dissipative conditions (as they would be in conflict with the horizontal arrows). The authors are encouraged to reconsider this graphical aspects (if appropriate);

We modified the figure to enhance its readability. In particular, arrows are now consistently used to represent net currents. Their thickness is proportional to their magnitude in the steady state regime. The new section Ic in the Supplementary Information explicitly shows that the net current flowing between M and M* has to be twice as large as the others.

(8) In the text associated to figure 4 the authors are encouraged to explicitly refer in the text to the dotted lines (b) and (c), because at first it is difficult to relate the description made in the text to a specific point of fig. 4a. Making the dotted lines more evident (extended) in fig 4 a may also improve the readability of the figure and the associated text. Also, it would be advisable to change/adjust the label of figures (b) and (c), as the concentration panels seem not referenced at first; this would help also the referencing to the concentration panels in the main text;

The figure has been improved accordingly.

(9) Please be more specific when stating that it was possible to reproduce "bottom left part of figure 4a";

We elaborated the sentence to make it more specific.

(10) After eq. 2, please consider if "inequality" is appropriate (equality?);

The word has been changed.

(11) At the end of page 2, please consider if "reaches" is appropriate (reached?);

The word has been changed.

(12) Please consider if the use of ""burns" chemical work" at page 3 is appropriate.

The term has been substituted with "consumes".

Reviewer #2 (Remarks to the Author):

In this paper the authors seek to develop a thermodynamic framework for analyzing the performance of chemical reaction networks (CRN) that accomplish either of two useful tasks: the accumulation of high energy molecules (energy storage, or ES), or the extraction of these molecules (dissipative synthesis, or DS).

In either case, the performance is characterized by efficiency, defined to be the ratio of a desired output to an expended input. For ES, the output is the nonequilibrium free energy [curly G], which begins at zero in equilibrium and grows until it reaches a plateau as the concentration of high energy molecules saturates. The input is chemical work, calculated for the reaction (in this case, $F \rightarrow W$) that generates and then sustains the nonequilibrium steady state. Since [curly G] saturates while chemical work grows linearly (asymptotically) with time, the efficiency decays to zero with time, which simply reflects that once the steady state is reached, the continual reaction $F \rightarrow W$ brings no added benefits, but merely maintains a nonequilibrium concentration of high energy molecules.

For DS, the output is the combined extraction of high energy molecules and injection of low energy ones, and the input is again chemical work. This situation is an example of free energy transduction, and efficiency is thus naturally defined in terms of output chemical work over input chemical work.

The analysis in this paper is correct, and the definition of efficiency in the ES case seems novel. But since this definition gives an efficiency that decays with time, it is not clear why this definition is a good one or why we should expect it to influence thinking in the field. Yes, η_{es} (equation 2) has a value between 0 and 1 in the ES regime, but why should we expect this to be a useful measure of performance?

In the DS case the definition of efficiency η_{ds} (equation 5) is quite reasonable, but in this case the efficiency is simply a ratio of output chemical work to input chemical work. This would seem to be a more or less "obvious" definition of efficiency for free energy transduction, which is the situation in which a downhill chemical reaction drives another reaction uphill. Admittedly, I can provide no reference where I have seen this definition used.

In short, while I find the analysis to be correct, I am not entirely convinced that the authors have developed a novel formalism that is likely to prove useful in the performance analysis of CRNs. To make a stronger case, the authors should address the following questions. For ES, how is their definition of efficiency useful, particularly as it decays to zero with time?

We agree with the referee that the relevance of our definitions was not sufficiently discussed. We now do so in the revised version (see red paragraph below Fig. 4). We hope that this paragraph makes it clear that the interesting features of the efficiency as a function of time lie in its short and medium time behavior. The long time convergence to zero simply reflects the fact that maintenance phase produces no storage and is thus purely dissipative.

We believe our performance measures to be insightful as they offer a quantitative optimization principle which was missing in the literature and may drive the design of new experiments.

For DS, does their definition differ from the ratio of output chemical work to input chemical work, in a straightforward example of free energy transduction?

There is indeed significant literature dealing with energy transduction. A leading figure in this field has been Terrell L. Hill, who made extended use of the ratio of output power and input power to define efficiencies in chemical models (Ref. [12] in the revised paper). However, this literature is restricted to chemical reaction networks (CRNs) only involving pseudo first order steps. Indeed, these networks lead to simple linear dynamics that can be represented as a Markov process on a simple graph, and this literature crucially relies on this. Our approach instead applies to any linear or nonlinear CRN. The network used in the paper contains for instance second-order reactions. This is the reason why it cannot be represented as a simple graph and more advanced methods based on the stoichiometric matrix (detailed in the supplementary material) must be used.

We now emphasize this point in the paper (sentence in red at the end of the first column of the first page as well as in the last concluding paragraph).

REVIEWERS' COMMENTS:

Reviewer #1 (Remarks to the Author):

The authors have implemented the reviewer's suggestions, where appropriate, and the quality of the manuscript has increased. I recommend publication in the present form.

Reviewer #2 (Remarks to the Author):

The authors have adequately addressed the issues that I raised in my original report.

In particular, with respect to ES they now clarify that their proposed measure of efficiency is most meaningful in the short / medium term (e.g. during the charging phase or shortly after the maintenance phase has been reached) rather than in the long-time limit when that measure decays to zero.

With respect to DS, I did not earlier appreciate the point about pseudo first order steps. The authors are correct to note that their formalism goes beyond what is already in the literature on free energy transduction.

I recommend that the manuscript be accepted for publication in Nature Communications.